# Diagnosing peri-implant disease using the tongue as a 24/7 detector

J. Ritzer[1], T. Lühmann[1], C. Rode[2], M. Pein-Hackelbusch[3], I. Immohr[3], U. Schedler[4], T. Thiele[4], S. Stübinger[5], B.v. Rechenberg[5], J. Waser-Althaus[6], F. Schlottig[6], M. Merli[7], H. Dawe[7], M. Karpíšek[8], R. Wyrwa[2], M. Schnabelrauch[2] & L. Meinel[1]

Our ability of screening broad communities for clinically asymptomatic diseases critically drives population health. Sensory chewing gums are presented targeting the tongue as 24/7 detector allowing diagnosis by "anyone, anywhere, anytime". The chewing gum contains peptide sensors consisting of a protease cleavable linker in between a bitter substance and a microparticle. Matrix metalloproteinases in the oral cavity, as upregulated in peri-implant disease, specifically target the protease cleavable linker while chewing the gum, thereby generating bitterness for detection by the tongue. The peptide sensors prove significant success in discriminating saliva collected from patients with peri-implant disease versus clinically asymptomatic volunteers. Superior outcome is demonstrated over commercially available protease-based tests in saliva. "Anyone, anywhere, anytime" diagnostics are within reach for oral inflammation. Expanding this platform technology to other diseases in the future features this diagnostic as a massive screening tool potentially maximizing impact on population health.

[1] Institute for Pharmacy and Food Chemistry, Universität Würzburg, Am Hubland, 97074 Würzburg, Germany. [2] Biomaterials Department, Innovent e.V., Prüssingstraße 27B, 07745 Jena, Germany. [3] Institute for Pharmaceutics, Universität Düsseldorf, Universitätsstraße 1, 40225 Düsseldorf, Germany. [4] PolyAn GmbH, Rudolf-Baschant-Straße 2, 13086 Berlin, Germany. [5] Musculoskeletal Research Unit, Center for Applied Biotechnology and Molecular Medicine, Universität Zürich, Winterthurerstrasse 270, 8057 Zurich, Switzerland. [6] Thommen Medical AG, Neckarsulmstrasse 28, 2540 Grenchen, Switzerland. [7] Indent—International Dental Research and Education srl, Via Settembrini 17/o, 47923 Rimini, Italy. [8] BioVendor—Laboratorni medicina AS and Department of Human Pharmacology and Toxicology, University of Veterinary and Pharmaceutical Sciences, Palackého 1-3, 61242 Brno, Czech Republic. J. Ritzer and T. Lühmann contributed equally to the work. Correspondence and requests for materials should be addressed to L.M. (email: lorenz.meinel@uni-wuerzburg.de)

Modern medicine frequently demands elaborate technical apparatus for diagnosis, hospital wards, and specialists largely unavailable to community and administrative medicine. Cheap diagnostic techniques based on easy to understand decision rules complement specialist approaches with widespread impact[1, 2]. Accordingly, we developed diagnostic chewing gums, using ones tongue as 24/7 available detector. These chewing gums bioresponsively develop a taste through enzymatic activity in the oral–nasal cavities. As proof of concept, we target the diagnosis of inflammatory condition in peri-implant diseased patients.

Sensor design was driven by physiological insights on taste. Structurally diverse molecules are bitter tasting, including polyphenols, flavonoids, or methylxanthines (caffeine)[3]. The main function of sensing bitter taste is linked to the detection of poisonous substances[4]. This would readily explain why bitter taste is so sensitively recognized with some molecules being recognized in the nanomolar range[4]. The human tongue senses bitterness of denatonium down to concentrations of 10 nM with consumer products containing as little as 30–100 ppm[5]. We translated these insights into design requirements of the diagnostic sensor. First, the large (uncleaved) intact sensor was designed for water insolubility and tastelessness. Second and in disease, the sensor is specifically cleaved by disease-induced matrix metalloproteinases in the patient saliva. This cleavage results in low molecular weight, water soluble, and bitter substances and it is for this bitterness that the patients are alarmed by recognizing a strong taste. The first requirement was met by deploying protease cleavable linker (PCL) inter-positioned between a microparticle and a bitter substance (this fully assembled setup is referred to as the sensor; Fig. 1a). Regarding the second requirement, we deployed matrix metalloproteinases (MMP) as present in peri-implant disease (mucositis or

peri-implantitis)[6–8]. Mammalian collagenases, particularly MMP-1, MMP-8, and MMP-9, are upregulated in periodontitis, gingivitis, and peri-implant disease[9–13]. These proteases cleave collagens at a common consensus sequence which was used for our peptide sensors (GPQG~IAGQ with ~ indicating the cleavage site)[14–17].

We designed two peptide sensor generations (Fig. 1a). MMP cleavage of the first-generation peptide sensor resulted in a weakly bitter peptide denatonium fragment with a free C-terminus (Fig. 1b). As of this free C-terminus, the peptide denatonium fragment was not a substrate for salivary aminopeptidases. Therefore, further digestion was prevented as a prerequisite to boost bitterness through reduction of the molecular weight. Consequently, we re-engineered the sensor such that the resulting peptide denatonium fragment was a substrate for salivary aminopeptidases, resulting in the second-generation peptide sensors (Fig. 1b).

We approached the challenges of modifying bitter denatonium without losing taste properties, identifying PCLs, and synthesizing these between the denatonium derivate and a microparticle. The resulting sensor was characterized in vitro for MMP-8 cleavage rate as function of the sensor's molecular structure, the sensor's selectivity against other MMPs, and the bitterness of its cleavage fragments. The sensor was successfully formulated into a chewing gum to provide a convenient diagnostic for patients. Finally, the sensor's functionality was tested in a study, incubating it in saliva collected from patients with peri-implant disease (defined as mucositis or peri-implantitis) vs. healthy volunteers (asymptomatic patients with at least one dental implant) and compared outcome to two commercially available tests, one chair-side test measuring MMP-8 activity from sulcus fluid sampled from the gingival pocket of the implant site and one protease activity test measuring in saliva. Thereby, the "anyone,

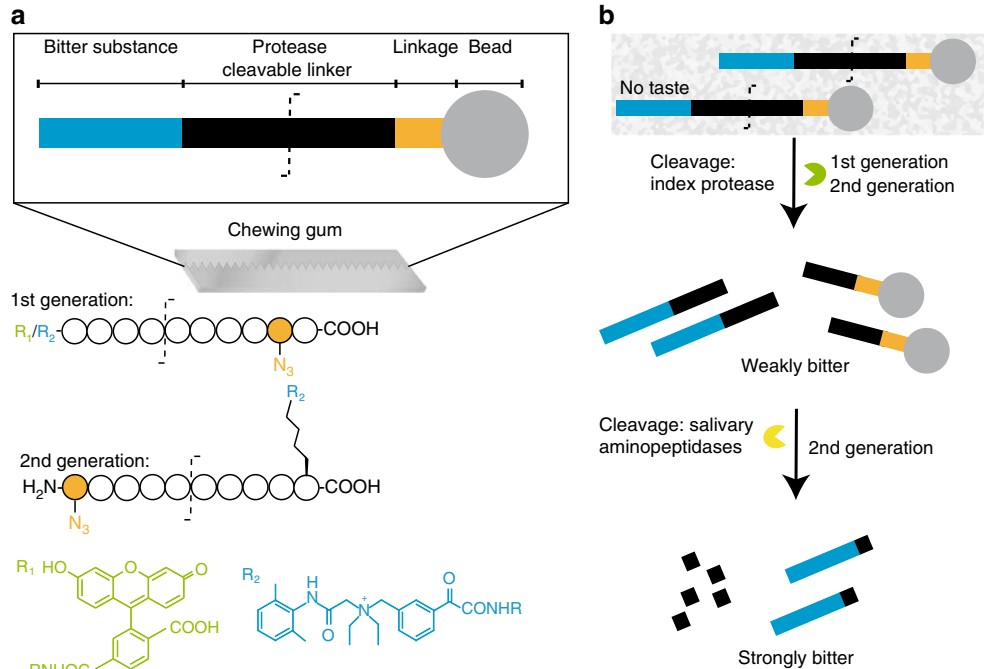

**Fig. 1** Cartoon outlining the functionality of the sensors. **a** Particle-bound protease cleavable linkers (PCL) with denatonium carboxylate derivate and schematic presentation of the PCL denatonium conjugates. First-generation sensors had the rests R₁ carboxyfluorescein, R₂ denatonium carboxylate) attached to the PCL's N-terminus. Second-generation sensors had the denatonium carboxylate attached to the ε−amino group of a C-terminal lysine. **b** Salivary MMPs cleave the sensor specifically, thereby creating a peptide denatonium fragment with a weak bitter taste. Second-generation sensors were designed such that this peptide denatonium fragment was substrate for salivary aminopeptidases resulting in further degradation of the peptide. This additional reduction of size of the peptide denatonium fragment increases bitterness. The *orange-filled* amino acid is azido alanine

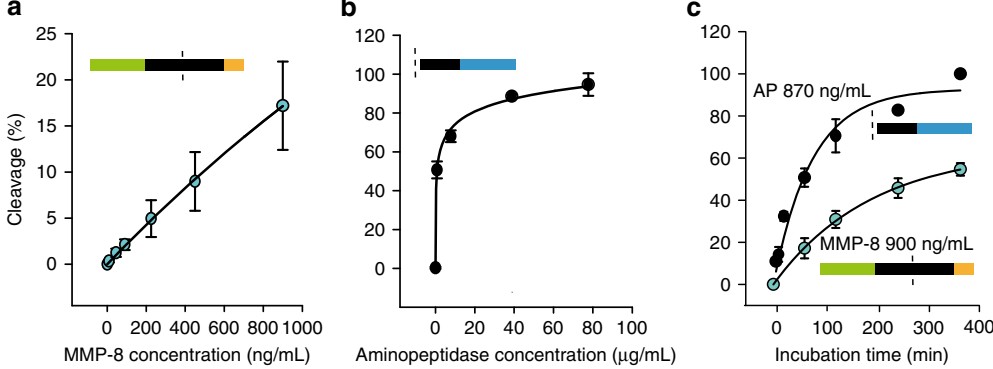

**Fig. 2** Concentration and incubation time-dependent cleavage of the PCL denatonium or PCL carboxyfluorescein conjugates. **a** Cleavage (%) of the PCL (Cf-CLAN₃Q in Supplementary Table 1; Fig. 1a) as a function of MMP-8 concentration within 60 min. $n = 10$. **b** cleavage (%) of the peptide denatonium fragment (i.e., IAGQK-De in Supplementary Table 1; Fig. 1b) as a function of aminopeptidase concentration within 60 min. $n = 5$. **c** Impact of incubation time on cleavage (%) of the PCL using 900 ng/mL MMP-8 $n = 8$, and on the peptide denatonium fragment using 870 ng/mL aminopeptidase $n = 5$. All data are presented as mean ± s.d. and each experiment was repeated at least once

anywhere, anytime" diagnostics for oral inflammation is presented, spanning from basic research to clinical translation.

## Results

**Design and performance of PCL denatonium conjugates.** A denatonium derivate was developed with a free carboxyl group for coupling to the PCL. For that, we synthesized a benzyl ring-substituted denatonium derivative (i.e., N-(3-carboxybenzyl)-2-(2,6-dimethylphenylamino)-N,N-diethyl-2-oxoethanaminium bromide) (Supplementary Figs. 1–4). Intermediates and products were characterized by ¹H-NMR, ¹³C-NMR, ¹³C-DEPT-NMR, ESI-MS, and ATR-IR (Supplementary Figs. 5–9) or HPLC and LC-MS (Supplementary Fig. 10). The product (referred to as denatonium carboxylate) was directly coupled to the N-terminus of the PCL. The PCLs had an unnatural azido alanine at their C-terminus through which the PCL was coupled to ethinyl-functionalized polymethylmethacrylate (PMMA) microparticles through the copper(I) catalyzed variant of the azide-alkyne Huisgen cycloaddition[18–20]. These peptide sensors were referred to as the first generation (Fig. 1). All synthesized peptide sensors are detailed (Supplementary Table 1; Supplementary Fig. 11).

MMP-8 salivary concentrations in healthy subjects have been reported at about 130 ng/mL. Patients with gingivitis or periodontitis had levels of 200 or 314 ng/mL[10]. Other studies assessed MMP-8 concentrations in gingival fluid at 100, 470, or 1850 ng/mL in healthy subjects and patients with gingivitis or periodontitis, respectively[21]. The first-generation PCL was tested for MMP-8 sensitivity (Fig. 2a, c), and cleavage of the resulting peptide denatonium fragment by salivary aminopeptidases (Figs. 1, 2b, c). These PCLs resulted in tetrapeptide cleavage products consisting of Cf-GPQG or D.-GPQG (Supplementary Figs. 12, 13). Control PCLs (exchange I5Y or Q3L) for the selected sequence (ID #1b) were synthesized and were not cleaved by MMP-8 up to 12 h (Supplementary Table 1; ID #3 and ID #4).

Moving from first-generation PCLs to the second generation—by inter-changing the azido alanine and denatonium from the first-generation sensors—increased the generated bitterness. The second generation resulted in peptide denatonium fragments with free N-terminus rendering substrates for salivary aminopeptidase. The first-generation PCL was not a substrate for aminopeptidase (AP) (Supplementary Fig. 13). In contrast, the second-generation peptide denatonium fragment was rapidly processed (Fig. 2b) resulting in 80% cleavage within 250 min when exposed to 870 ng/mL enzyme (Fig. 2c). For comparison, we measured

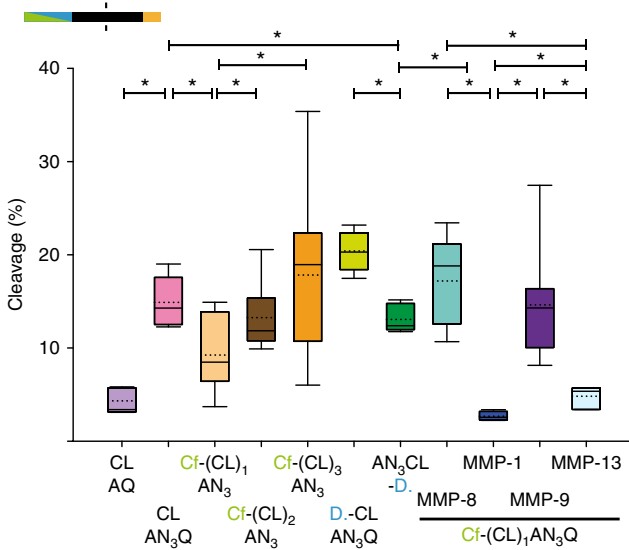

**Fig. 3** Cleavage (%) of PCL conjugates (incubation for 60 min). Detailed information on the tested peptides is provided in Supplementary Table 1. CL protease cleavable linker with the sequence GPQG~IAGQ with ~ indicating the MMP cleavage site. Amino-acid variations at the flanks of the PCL are given in one letter amino-acid code; AN₃ azido alanine, Cf carboxyfluorescein, D denatonium carboxylate and (CL)₂, and (CL)₃ are two and three consecutive repetitions of the PCL. MMP concentrations were 900 ng/mL and all experiments were performed with $n ≥ 5$. CLAQ $n = 9$, CLAN₃Q $n = 5$, Cf-(CL)₁AN₃ $n = 10$, Cf-(CL)₂AN₃ $n = 10$, Cf-(CL)₃AN₃ $n = 9$, D-CLAN₃Q $n = 9$, AN₃CL-D. $n = 6$, Cf-(CL)₁-AN₃Q (MMP-8) $n = 13$, Cf-(CL)₁-AN₃Q (MMP-1) $n = 7$, Cf-(CL)₁-AN₃Q (MMP-9) $n = 7$, Cf-(CL)₁-AN₃Q (MMP-13) $n = 5$. Statistical differences among means were assessed by pairwise comparison using a t-test and indicated by asterisks ($p < 0.05$). Box-and-whisker plots show the median (*solid line*) and mean (*dotted line*) and *whiskers* indicate the upper and lower 25%, respectively

aminopeptidases levels in saliva collected from healthy volunteers and from patients with an inflammation in the oral cavity resulting in identical average concentrations of 183 ± 155 ($n = 36$) and 169 ± 143 ng/mL ($n = 15$), respectively (Supplementary Fig. 14). The impact of incubation time on cleavage was determined for 900 ng/mL MMP-8, leveling off at >50% after 360 min (Fig. 2c).

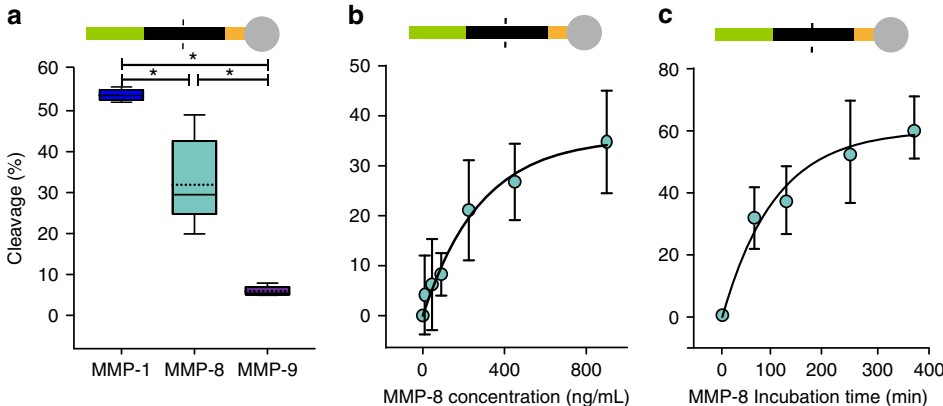

**Fig. 4** Cleavage of peptide sensors consisting of microparticle, PCL, and carboxyfluorescein. **a** Cleavage (%) of the sensor by MMP-1 $n = 5$, MMP-8 $n = 11$, and MMP-9 $n = 5$. The PCL in the sensor was most sensitive to MMP-1 and to a lesser extent to MMP-8, which in return cleaved the sensor better than MMP-9. Statistical differences among means were assessed by pairwise comparison using a $t$-test and indicated by *asterisks* ($p < 0.05$). Box-and-whisker plots show the median (*solid line*) and mean (*dotted line*) and *whiskers* indicate the upper and lower 25%, respectively. **b** Sensor cleavage (%) as a function of MMP-8 concentration (incubation for 60 min) $n = 7$ and **c** over time at a concentration of 900 ng/mL $n = 9$. All data are presented as mean ± s.d. and each experiment was repeated at least once

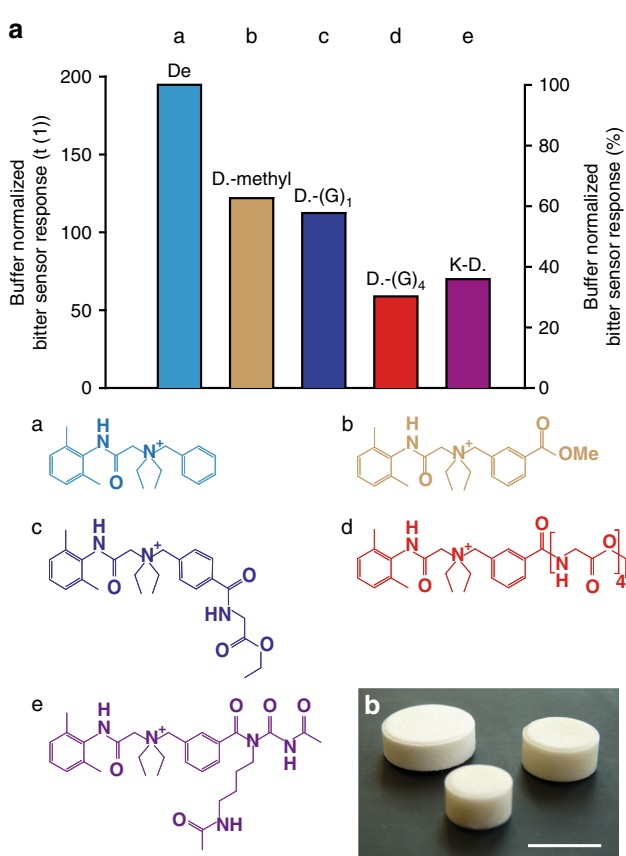

**Fig. 5** Potentiometric analysis of bitterness in Euclidian distances [t(1)] (see Supplementary Fig. 12 for details). **a** Bitterness response of denatonium (De; structure a), structure b (D-methyl), structure c (D-(G)₁), structure d (D-(G)₄), and structure e (K-D). Values were normalized to plain buffer responses. All experiments were performed with $n = 3$. **b** Prototype of the compressed chewing gums. *Scale bar* length equals 1 mm

We gained further insight on MMP-8 efficacy as a function of modifications on the PCL and its flanking sites (Fig. 3; detailed information on the PCLs and their modifications is provided in Supplementary Table 1; Supplementary Fig. 11). For example,

replacing alanine by azido alanine in the C-terminal region significantly improved cleavage ($p < 0.001$, $t$-test). Repeated assembly of PCLs increased cleavage from one to two, or one to three repeated units, respectively ($p < 0.05$, $t$-test). N-terminal attachment of the denatonium carboxylate ($p < 0.001$, $t$-test) but not of carboxyfluorescein significantly increased cleavage as compared to the non-attached control. Cleavage of the PCL denatonium conjugate was significantly better than for the PCL carboxyfluorescein conjugate (ID #1b; $p < 0.001$, $t$-test). This carboxyfluorescein conjugate (Figs 2a and 3) was further characterized for MMP selectivity (Fig. 3). MMP promiscuity of the PCL carboxyfluorescein conjugate (ID #1b) was assessed. Cleavage was best for MMP-8 and MMP-9 and significantly less for MMP-13 ($p < 0.05$, $t$-test), which cleaved better than MMP-1 ($p < 0.05$, $t$-test) (Fig. 3), similar to previous reports[16].

**Characterization of the peptide sensors.** At this point we turned from studies on the soluble PCL conjugates to assessments of the fully assembled, water insoluble sensor. Alkyne group-functionalized PMMA microparticles reacted rapidly with the azido alanine of the PCL carboxylate conjugate by bio-orthogonal copper-catalyzed 1,3-dipolar azide-alkyne cycloaddition ("click-reaction"). The reaction depended on Cu⁺ as indicated by fluorescence microscopy (Supplementary Fig. 15A) and flow cytometry analysis (Supplementary Fig. 15B, C). PCL MMP sensitivity changed from the soluble PCL conjugates to the fully assembled sensor. Whereas the soluble PCL fluorescein conjugate (ID #1b) alone had a sensitivity as of MMP-8 = MMP-9 > MMP-13 > MMP-1, the identical PCL inserted into the sensor was most sensitive to MMP-1 and to a lesser extent to MMP-8 ($p < 0.001$, $t$-test), which in return cleaved the sensor better than MMP-9 ($p < 0.001$, $t$-test; Fig. 4a). In spite of the highest sensor sensitivity to MMP-1 (Fig. 4a), we decided to continue our in vitro study using MMP-8 to allow comparison of the insoluble sensor results to the experiments conducted on the soluble PCL conjugates alone. Sensor cleavage in response to MMP-8 concentrations followed an exponential pattern (Fig. 4b), resulting in overall cleavage of about 60% (Fig. 4c), which is similar to what was observed for the PCL carboxyfluorescein conjugate alone (Fig. 2c). This indicated that MMP-8 cleavage was not substantially impacted by the presence of the microparticle—the opposite was observed for MMP-9, showing cleavage for the PCL

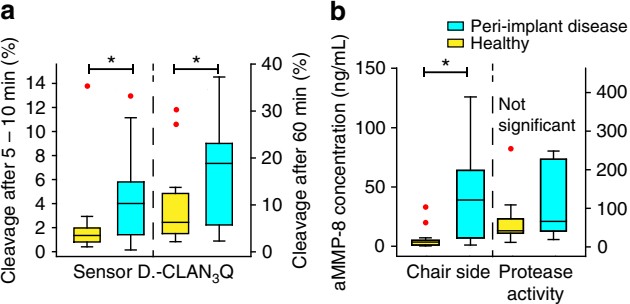

**Fig. 6** Sensor performance in comparison to commercially available MMP diagnostics in saliva or sulcus fluid collected from patients with peri-implant disease (defined as mucositis or peri-implantitis; $n = 19$) and healthy control ($n = 14$). **a** Cleavage of the sensor in % after 5–10 and 60 min of incubation in saliva. **b** Active MMP-8 (aMMP-8) in sulcus fluid (chair-side test) and as measured from saliva (QuickZyme MMP-8 activity test). Box-and-whisker plots show the median (*solid line*), the *box* contains the middle 50% of values and the *whiskers* show the lower and upper 25% of the data, respectively. Outliers (not used for the analysis) are shown as *red dots*. *Asterisks* indicate statistical difference as assessed by a *t*-test ($p < 0.05$)

alone (Fig. 3) but hardly any cleavage when the same PCL was integrated into the sensor (Fig. 4a).

**Quantification of bitterness**. Bitterness was quantified with a potentiometric setup followed by multi-variate analysis and normalized to buffer response (Fig. 5a, Supplementary Fig. 16, 17; Supplementary Table 2)[22]. Unmodified denatonium was tested for comparison at concentrations of 0.5 mM (50,000-fold above bitterness as recognition thresholds reported for humans) and 0.1 mM (Supplementary Fig. 16). Free carboxyl or amino groups impacted the potentiometric read-out, which is why these groups were masked by methyl ester (Supplementary Figs. 2, 7), glycine amide (Supplementary Figs. 4, 9), tetra-glycine amide (4G; Supplementary Fig. 11; ID #6), and an acetylated lysine derivative (Supplementary Fig. 11; ID #7). Increasing the size of the introduced side group decreased bitterness. For example, the tetra-glycine peptide denatonium fragment (Fig. 5a(d))—this fragment was synthesized to approximate the taste properties of the peptide denatonium fragment before AP digest—was substantially less bitter than the mono-glycine denatonium fragment (Fig. 5a(c)). The ultimate denatonium fragment of the peptide sensor after MMP cleavage and followed by AP digest of the resulting peptide denatonium fragment was synthesized (Supplementary Fig. 11, ID #7). Bitterness of this fragment was about 40% of denatonium itself at identical concentrations of 0.5 mM.

**Manufacturing of peptide sensor-loaded chewing gums**. We also aimed at demonstrating the general feasibility of integrating the sensors into chewing gums (Fig. 5b). For that, fluorescence-labeled PMMA microparticles were blended into a compressible chewing gum base and tableted. Blending of carboxyfluorescein-decorated PMMA particles with diameters of 2, 5, or 17 μm was homogenous. Based on these manufacturing characteristics, we selected 9 μm particles for the assembly of the sensors. The chewing gum was easily compressed—three different chewing gum bases were characterized (Supplementary Table 3)—on an eccentric tableting machine and the resulting force-displacement curves indicated proper compression performance (Supplementary Fig. 18). The resulting chewing gums had acceptable break-forces up to 100 N.

**Sensor performance in saliva**. We tested our peptide sensor (sensor's PCL ID #1c) in saliva in comparison to a commercially available dental chair-side test probing for activated MMP-8 and measuring from sulcus fluid[23] as well as in comparison to the QuickZyme Human MMP-8 activity assay in saliva, which was also responding to MMP-1, MMP-9, and MMP-13. The dentists monitored the clinical status and collected sulcus fluid from the implant site as well as saliva from both healthy volunteers ($n = 14$; defined as patients with a clinically asymptomatic dental implant (s); i.e., not having signs and symptoms of peri-implant disease) as well as from patients with peri-implant disease (defined as mucositis/peri-implantitis; $n = 19$) (Fig. 6; Supplementary Fig. 19). Cleavage of the sensor was significantly different among groups after incubation for 5–10 min with $1.5 \pm 0.8\%$ and $4.2 \pm 3.34\%$ ($p < 0.01$, *t*-test) and also after an incubation time of 60 min with $7.6 \pm 4.4\%$ and $17.1 \pm 11.1\%$ ($p < 0.01$, *t*-test) for healthy vs. diseased subjects, respectively. The chair-side test—probing from sulcus fluid of the affected site and not from saliva as our sensor—significantly discriminated among healthy and diseased subjects resulting in $3.2 \pm 2.4$ and $44.2 \pm 42.3$ ng/mL, respectively ($p < 0.01$, *t*-test). In contrast, the QuickZyme protease activity test—probing from saliva like our sensor—did not significantly discriminate among groups ($52.1 \pm 31.2$ and $109 \pm 101$ ng/mL, $p > 0.05$, *t*-test).

We also recorded the presence of periodontitis in the trial (Supplementary Fig. 20). None of the tests discriminated periodontitis patients from healthy volunteers.

## Discussion

"Anyone, anywhere, anytime" diagnostics were developed for peri-implant disease. The sensors responded to MMPs and provided proof of concept in statistically differentiating patients with peri-implant disease from healthy volunteers (having at least one asymptomatic dental implant).

One of the challenges was the generation of sufficiently bitter fragments following MMP cleavage of the sensor within reasonable (chewing) times. This challenge was solved by designing PCL degrading through a two-step mechanism. The initial (specific) cleavage of the water insoluble sensor was by MMPs resulting in a water soluble peptide denatonium fragment with a free N-terminus. As of the free N-terminus, the fragment was substrate to salivary aminopeptidases further digesting from the N-terminus, thereby reducing the size of the fragment and its bitterness (Figs. 3, 5). Thereby, the bitterness was bioresponsively generated in response to elevated MMP in saliva and patients can be alarmed of this potential sequalae. These platforms are not intended to rival sophisticated antibody-based screening tests let alone advanced next-generation sequencing technologies in terms of sensitivity or selectivity. However, they are designed to function in absence of complex machinery and/or expert knowledge for read-out interpretation in an effort to give a rapid, easy to interpret heads-up to anyone, anytime, and anywhere, without the need for external power supply or complex machinery/expert knowledge. Based on our study on saliva samples, the sensor indeed rivaled a chair-side tests in terms of discriminative potential among healthy and diseased subjects with the advantage of the sensor to measure from saliva (and not from sulcus fluid as the chair-side test) and instantaneous read-out (within 10 min and removing the requirement of the chair-side test for sending the sulcus fluid to a specialized lab for analysis; Fig. 6). In addition, the sensor performed better than the commercially available QuickZyme test measuring MMP proteases from saliva. Calculated concentrations of the cleaved denatonium derivative in saliva exceeded the detection limit of the tongue (10 nM) by a factor of ~40[5]. This result indicated that

the large amounts of denatonium generated through cleavage of the sensor is easily sensed by the human tongue.

The chewing gum diagnostic introduces several improvements. First, it is providing rapid read-out within a few minutes. Second, the sensor is featuring a diagnosis from saliva thereby removing the need to collect from sites accessible to a dentist only (sulcus fluid). Third, expert knowledge for interpretation is not required while finally the diagnostic can be used anywhere. Therefore, the data provided evidence that the complex kits used today may be complemented or even replaced by much simpler, reliable chewing gum diagnostics. Future applications include the use in a dentist's office or personal use by anyone, anywhere.

We developed the sensor to detect inflammation in the oral cavity, which is why we pooled the data for patients with mucositis or peri-implantitis (Fig. 6). Nevertheless we also reported the outcome for each indication (Supplementary Fig. 19). In spite of the fact that the sensor proofed successful in differentiating mucositis and peri-implantitis, the clinical relevance of this finding might be low. Patients will likely encounter difficulties in reporting different levels of taste generation (i.e., mucositis inducing a weaker taste than per-implantitis). Nevertheless, other formats of this diagnostic (e.g., using optical and not gustatory readouts) might use the sensor's capability in distinguishing among these indications.

The sensor was not selective to individual MMPs and responded to sets of proteases (e.g., MMP-1, -8, and -9) as upregulated in the saliva in patients suffering from peri-implant disease[9, 13] as a result of bacterial infection[24].

The example provided here for MMPs is the first prototype of this platform technology, featuring the "anyone, anywhere, anytime" diagnostic paradigm and as demonstrated for peri-implant disease. It is for this easy use that this diagnostic may help to reduce consequences of peri-implant disease through close-meshed self-observation, including degradation of maxillary/mandibular bone and/or implant loss[25–27]. Future diagnostics are easily accessible through this platform technology. By replacing the MMP-sensitive PCL of the sensor by other PCLs, new diagnostics can be generated. We currently have sensors in development responding to Aureolysin (alarming for infectious *Staphylococcus aureus* in the nasal/oral cavities with a potential use in screening patients before hospital admission to reduce bacterial challenges) or another sensor stratifying sore throat patients into those with influenza or *Streptococcus pyogenes* (in an effort to reduce the unjustified prescription of antibiotics while providing diagnostic information on the possible presence of problematic *S. pyogenes*).

Several restrictions apply to the study. Taste sense is dependent on demographic variables as ethnicity, sex, age as well as on experience, and partially defined by genetics with a large inter-individual variability[28]. However, sensitivity to denatonium (Bitrex) is quite conserved, which is why the molecule is used in consumer products to prevent accidental poisoning. It is for this reason and to expand on the electronic tongue setup used in this study (Fig. 5)[29, 30] that validated clinical studies are required in the future with "bitterness" as clinical endpoint. Finally, sensor sensitivity can be improved. For example, the mechanisms of recognition and cleavage of the native collagen triple helix by MMPs has been recently deciphered and these insights can drive future PCL designs. For example, combining the current PCL with the interaction interphase of the non-catalytical hemopexin domain of collagen is expected to substantially increase both, the cleavage rate and the cleavage kinetics[31]. However, proof of concept was provided in discriminating patients from healthy volunteers (Fig. 6) within reasonable (chewing compatible) times (<10 min), the overall cleavage was only <5% in patients

and <1.5% in healthy volunteers. We estimated based on these cleavage percentages that ~10 mg sensor are sufficient per chewing gum to yield reasonable taste responses indicating the potential of this platform in future settings.

The next developmental step is testing the chewing gum in humans. Thereby, the proof of concept provided here in saliva samples will be pushed to its ultimate application in an effort to demonstrate the ability to discriminate diseased from healthy groups through taste generation. We are currently assessing the (GMP) manufacturing, analytical, and (GCP) clinical requirements to conduct this next step.

In conclusion, chewing gums were developed featuring bioresponsive, protease triggered release of a flavoring substance in response to target proteases. Anyplace, anyhow, anywhere diagnostics of peri-implant disease come in reach, using the tongue as a 24/7 available detector. These systems are independent of complex machinery, expert knowledge, or power supply. The study on collected saliva and sulcus fluid samples indicated equal or better performances of the sensor in comparison to commercially available test systems. The study lacks a clinical trial at this stage. Future clinical trials should aim at pushing the proof of concept provided here into patient recorded outcome measures.

## Methods

**Synthesis of denatonium derivatives**. A method for synthesizing denatonium derivatives allowing coupling reactions was developed. For that, a carboxyl group was introduced by using ring-substituted (halomethyl) phenyl derivatives with lidocaine as educts. Quaternization reactions of lidocaine did not require solvents and proceeded at room temperature. Detailed methods and the synthesis of denatonium derivatives coupled to amino acids as well as the synthesis of acetylated diaminoethyl denatonium can be found in the Supplementary Methods.

**Solid-phase peptide synthesis**. All PCL sequences and other peptides were synthesized in-house using standard Fmoc solid-phase peptide synthesis and different resins. Peptides were analyzed and purified with MALDI, LC-MS, FPLC and HPLC as further described in the Supplementary Methods.

**MMP incubation**. Pro-MMPs (e.g., MMP-8, MMP-1, or MMP-9; used at 0.1 mg/mL and >100.0 mU/mg based on the manufacturer's information) were activated with p-aminophenylmercuric acetate (APMA) (8). For that, a 10 mM stock solution of APMA was freshly prepared in 0.1 M NaOH and a 10:1 (v/v) ratio of the respective proenzyme and the APMA solution (MMP:APMA) was incubated for 3 h at 37 °C. The activity was confirmed using the MMP-8 detection kit (QuickZyme Biosciences, Leiden, the Netherlands). A stock solution (90 μg/mL) of the activated MMP was prepared (used for cleavage studies on the PCL or the fully assembled microparticle-PCL-tastant setup (sensor), in MMP-8 buffer (200 mM NaCl, 50 mM Tris-HCl, 5 mM CaCl₂, 1 μM ZnCl₂, 0.05% Brij 35 at pH 6.8–7.0). All MMP experiments are described in detail in the Supplementary Methods.

**Aminopeptidase incubation**. Leucine aminopeptidase (AP; Mw = 280 kDa) from porcine kidney dissolved in AP buffer (50 mM Tris-HCl, 1 mM CaCl₂, and 150 mM NaCl at pH 7.0) was used. L-leucine-p-nitroanilide (0.125 mg/mL) was used as control substrate. For AP digestion, 1 mg of purified lyophilized peptide was diluted in 2 mL AP buffer resulting in a final concentration of 0.5 mg/mL substrate. To evaluate the impact of AP concentration on peptide cleavage, AP was added at concentrations of 0 (negative control), 0.87, 8.7, 39, and 78 μg/mL and the samples were incubated at 37 °C for 1 h. For analysis of the AP incubation time, an AP concentration of 0.87 μg/mL was used and analyzed after 10, 20 min and 1, 2, 3, and 6 h. To stop the enzyme activity, samples were kept at 95 °C for 15 min. Afterwards, HPLC analyses were performed (as described in the Supplementary Information) to analyze the relative decrease of the main peak as compared to the negative control. Further details are given in the Supplementary Methods.

**Peptide-coupling to beads (sensor)**. For the Cu(I) catalyzed cycloaddition reaction (click reaction), 5,6-carboxyfluorescein-coupled azido-peptides and custom-made, transparent PMMA beads carrying alkyne functional groups were used. These alkyne functional groups were covalently coupled onto the surface of the beads using a photoinitiated grafting procedure using propargylacrylamide as monomer and benzophenone as photoinitiator. Alkyne beads (1–2 mg) were coupled to the PCL in 100 μL Tris-buffer (50 mM, pH 7.5), 2 μL of a 1:1 (v/v)

mixture of 100 mM $Cu_2SO_4$, and 100 mM TCEP in water, 0.4 μL TBTA (5 mM in DMSO, tertiary butanol (1:4, v/v)), and 10 μL azido-peptide (4 mM). After incubation for 60 min, the beads were washed as described in the Supplementary Information. Negative controls were prepared under identical reaction conditions without $Cu_2SO_4$ and TCEP. Finally, the beads were resuspended in 500 μL water (LCMS grade) and analyzed by fluorescence microscopy and flow cytometry. For coupling the PCL to microparticles, Cu(I) catalyzed cycloaddition reaction was used. Details regarding fluorescence microscopy and flow cytometry experiments are in the Supplementary Methods.

**Electronic tongue measurements**. Taste evaluation was carried out with the potentiometric electronic taste sensing system TS-5000Z (Insent, Atsugi-Shi, Japan). Detailed methods can be found in the Supplementary Methods.

**Chewing gum compression**. For the production of chewing gums three directly compressible powders were used in combination with PMMA particles differing in bead size as described in detail in the Supplementary Methods.

**Study with patient saliva**. Volunteers were enrolled for a cross-sectional study. One group had peri-implant disease (defined as peri-implantitis or mucositis), the other group were healthy volunteers (asymptomatic patients with at least one dental implant). Unstimulated saliva was collected using an absorbent device. Sulcus fluid was collected from the peri-implant zone using standardized, commercially available MMP-8 collection strips and sent to Dentognostics GmbH (Jena, Germany) for analysis (referred to as chair-side test). Determination of total MMP activity in saliva samples was conducted using the human MMP-8 activity assay (QuickZyme, Leiden, the Netherlands) according to the manufacturer's instructions. Saliva samples from the cross-sectional study were incubated with D-CLAN₃Q (ID #1c) in MMP-8 buffer and cleavage was analyzed by HPLC. Further details are given in the Supplementary Methods.

**Statistical analysis**. Statistical analyses were performed using Sigma Plot version 13 (Systat Software, San Jose, CA) and Minitab version 17 (Minitab Inc., State College, PA). The two-sample $t$-test (two-sided) was used for pairwise comparison assuming equal variances of the two populations (fluids from peri-implant-diseased patients vs. control) and normal distribution (after $F$-test). Results from the study on saliva samples outside the 6σ band were classified as outliers and were not included into the analysis but are shown (Fig. 6). $P$-values <0.05 were considered statistically significant. The s.d. was used to quantify the reproducibility. For all box-and-whisker plots the box contains the middle 50% of values and the whiskers show the *lower* and *upper* 25% of the data, respectively.

**Data availability**. The data sets generated during and/or analyzed during the current study are available from the corresponding author on reasonable request.

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

## Acknowledgements

We acknowledge experimental help by Anna-Sophia Stecher and Katja Tschirner (for technical support with the PCL synthesis), Dominik Dolles (for technical support with the LC-MS experiments), Oliver Germershaus (University of Würzburg, now at the Fachhochschule Nordwestschweiz), Stefanie Lahmer and Michael Büchner (for MALDI-MS measurements), Alix Hillenhagen and Elisabeth Geiß (support with the chewing gum blending and tableting of the chewing gum), Norbert Steiger (collection of saliva from patients for AP quantification). Financial support was provided by the—European Union—Seventh Framework Program (Grant 314911-"STEP") and the Bundesministerium für Bildung und Forschung (German Federal Ministry of Education and Research; #13N13454) is gratefully acknowledged.

## Author contributions

C.R., R.W., and M.S. designed and performed the experiments regarding the bitter substance modification including analysis. L.M., T.L., and J.R. planned, performed, and analyzed all experiments regarding peptide synthesis, coupling, tableting, microscopy, flow cytometry as well as protease cleavage (MMP, aminopeptidase and

salivary incubation and analysis) experiments. The study aiming at saliva collection was planned by B.R., S.S., J.W.A., F.S., M.M., and H.D. and conducted by S.S. and M.M., I.I., and M.P.H. performed the electronic tongue experiments. M.K. conducted the QuickZyme Assay. U.S. and T.T. prepared the microbeads and conducted the analysis thereof. J.R., T.L., and L.M. wrote the manuscript and all authors reviewed the manuscript.

## Additional information

**Competing interests:** L.M., M.S., and F.S. have a patent application (WO2013131993) and state a possible conflict of interest. F.S. and J.W.-A. were or are full time associates of Thommen Medical AG and U.S. and T.T. are full time associates of PolyAn GmbH. All four authors declare a possible conflict of interest. The remaining authors declare no competing financial interests.

