## [Peer Review File · Nature Communications]

Reviewers' Comments:

Reviewer #1:

Remarks to the Author:

This manuscript showed very effective 24/7 biosensor to detect MMP8 for the fast diagnosis of peri-implant disease. The idea to utilize patients' tongues as sensing transducer is very practical with high novelty and experimental logic is clear.

However, there are some limits this scientific idea to be realized.

This sensing platform may be too dependent on patients conditions. Independency of sensor capability needs to be shown.

Based on normal sensing tools, as known so far, sensors consist of recognition part and signal generation (transducer) part. In here, MMP8 is chosen as a target for peri-implant disease. Tongue was utilized as signal generation part (Transducer) and bitter molecule-MMP cleavable linker-PMMA micro particle complex was employed as MMP recognition part. Saliva (aminopeptidase contained) was also utilized as signal enhancing reagent. Therefore, I think that, according to this sensing format, sensitivity of sensor (tongue) and signal enhancing capability (concentration of aminopeptidase in patients saliva) depends on patients. Therefore, this sensor inherently has big signal variations (in figure 6).

To overcome these ones, I'd like to recommend some more results are required.

1. I recommend author to confirm the concentration of aminopeptidase in clinical sample (used in figure 6) and break down capabilities are similar or not in the concentration ranges of aminopeptidase (Author showed similar result in figure 2B, however, this result did not show real one (in saliva, embedded in gum paste, and with PMMA micro particles).

2. I recommend the reference tastes or gums to normalize patients' tongue sensitivity.

3. To be alarming sensing tools in field, gray zone between positive and negative samples should be exist. I recommend author to decide gray zone and obtain sensitivity and selectivity based on blind test (with designed sensor and Quickzymes). I expect figure 6 could be more understandable if sensitivity and selectivity results are added).

In minor recommendation,

1. I think if you give easy names to various linkers, this manuscript would be more readable. 1st generation and 2nd generation and so many names in table (supplementary) are exist in manuscript.

2. Frankly, 1st generation linker may not be required in main manuscript. I think it would be more interesting when sensor works are focused.

Reviewer #2:

Remarks to the Author:

- 1) The authors show that there is some level of non-specific reactivity (with either MMP9 or MMP1, depending on the substrate, see figure 3 and 4) but it is not clear if this will have an impact on the specificity of the assay to detect peri-implant disease. The authors only comment on the biomarker role of MMP8 in this clinical context, but I am not sure if the other MMPs are also implicated.

- 2) I think that an important piece is missing, as there is no clinical testing of the gum in patients. The authors went a long way to design the probe and link it to denatonium, to study the different degrees of "bitterness" of the cleaved compounds, and to formulate a chewing gum with the right amount of probes, and then did not show the effectiveness of the gum itself, but instead only used the alternative version of the probe (linked to fluorescein) to show that it is cleaved with saliva from patients.

- 3) The head-to-head study in figure 6 is good, but it looks as if the probe is equally good as the

standard test already used, so if there is not an improvement on the specificity/sensitivity the authors should at least comment on how they think a test like this could be implemented clinically and help patients. The idea of doing the test autonomously at home is proposed, but in practical terms how would this work?.

Reviewer #3:

Remarks to the Author:

Title: Diagnosing peri-implant disease deploying the tongue as 24/7 detector

Authors: Ritzer et al.

The study of Ritzer et al. represents a new concept in diagnosing an oral condition referred to as peri-implant disease or peri-implantitis. This emerging disease is a condition that is increasing in all industrialized countries. Diagnosis and treatment of this condition is time-consuming and represents an increasing medical and economic burden. Early detection of this condition enables effective intervention and may contribute to the prevention of implant loss. The concept of self-diagnosis by using a sensory chewing gum is a novelty and will be greatly appreciated in the dental community.

The study comprises of a pre-clinical and a clinical part. The technical, i.e. the pre-clinical part of the study, has been described meticulously and has been tested in vitro. The device seems ready to be tested in humans.

Comments

1. The authors have tested their invention with saliva samples obtained from healthy subjects and patients with peri-implant disease, involving both mucositis and peri-implantitis. It would have been much better if they would have distinguished between mucositis and peri-implantitis. This would have been possible since the authors have selected patients with only mucositis and patients with peri-implantitis (Suppl. files, Clinical protocol).

2. MMP-8 is a collagen-degrading enzyme from polymorphonuclear leucocytes. These leucocytes are present in the oral cavity in both health and disease which is also shown in Figure 6. Why did the authors select subjects with peri-implant inflammation for this study and not also gingivitis and periodontitis patients? All these conditions will provide a MMP-8-based positive signal with the new invention.

3. Why did the authors tested saliva samples and why not have subjects chew on the experimental device? The title suggests that the authors have tested "the tongue as a 24/7 detector" but this is not what actually has been studied. Also, the abstract suggest that subjects have actually used the device (40-42).

4. The authors did not show that the device works clinically.

5. The corresponding author has filed several patents relating to diagnostic chewing gums (Patent number: 9526803, Publication number: 20150023879). This is not mentioned in the manuscript. One of these patents relates to oral inflammation in general and not to peri-implant disease in particular. Such a patent can be related to a conflict of interest.

On behalf of all authors, I like to express our great appreciation for the valuable comments made by the reviewers. We are providing the reviewer's feedback in the frame, followed by our comments.

Reviewer #1 (Remarks to the Author):

This manuscript showed very effective 24/7 biosensor to detect MMP8 for the fast diagnosis of peri-implant disease. The idea to utilize patients' tongues as sensing transducer is very practical with high novelty and experimental logic is clear.

However, there are some limits this scientific idea to be realized.

This sensing platform may be too dependent on patients conditions. Independency of sensor capability needs to be shown.

Based on normal sensing tools, as known so far, sensors consist of recognition part and signal generation (transducer) part. In here, MMP8 is chosen as a target for peri-implant disease. Tongue was utilized as signal generation part (Transducer) and bitter molecule-MMP cleavable linker- PMMA micro particle complex was employed as MMP recognition part. Saliva (aminopeptidase contained) was also utilized as signal enhancing reagent. Therefore, I think that, according to this sensing format, sensitivity of sensor (tongue) and signal enhancing capability (concentration of aminopeptidase in patients saliva) depends on patients. Therefore, this sensor inherently has big signal variations (in figure 6).

To overcome these ones, I'd like to recommend some more results are required.

1. I recommend author to confirm the concentration of aminopeptidase in clinical sample (used in figure 6) and break down capabilities are similar or not in the concentration ranges of aminopeptidase (Author showed similar result in figure 2B, however, this result did not show real one (in saliva, embedded in gum paste, and with PMMA micro particles)).

To this end, we have again collected saliva from healthy donors and patients with an oral inflammation and quantified the aminopeptidase (AP) levels according to the reviewer's advice.

→We are detailing this outcome in Figure S7 and pages 5 and 6 of the manuscript in its revised form.

2. I recommend the reference tastes or gums to normalize patients' tongue sensitivity.

3. To be alarming sensing tools in field, gray zone between positive and negative samples should be exist. I recommend author to decide gray zone and obtain sensitivity and selectivity based on blind test (with designed sensor and Quickzymes). I expect figure 6 could be more understandable if sensitivity and selectivity results are added).

Thank you very much for this valuable comment. Blind tests as suggested require testing in humans. We would not obtain ethical approval to test our chewing gums in humans as of our manufacturing restrictions at universities. Instead, we would need to transfer our manufacturing to dedicated sites working in proper quality regulated environments relating to or being under GMP. We do not have funding to initiate such extramural fee for service work at present. Furthermore, we would need to use

validated (GMP regulated) release tests for the product, which can only be conducted at dedicated and regulatory audited sites (which are fee for service labs). In conclusion, neither the manufacturing nor the analytical GMP conductions are available to us. Conducting these studies would approximately cost 0.5 million USD. In addition, the subsequent clinical (GCP) study would cost about 1.5 million US Dollar. All these efforts would take approximately 2 years before we could provide these data sets and providing the financial resources would already be available. We apologize for providing this detailed information on our financial constraints to the reviewer but believe it is important to communicate our restrictions to overcome this hurdle. However, we believe that our thorough testing using the electronic tongue - a broadly accepted industrial method frequently used in the food and pharmaceutical industry - and detailing the sensor's performance with respect to cleavage in saliva provided an adequate proof of concept for this diagnostic platform.

What was missing is the correlation of the bitter cleavage product (denatonium) generated by disease-related proteases in patient saliva to the same concentrations of denatonium measured in buffer with the electronic tongue. This can be done indirectly (a direct assessment of a bitter substance with the electronic tongue is not feasible in saliva. The electronic tongue is a set of potentiometric sensors which are disturbed in a complex matrix such as saliva). However, we are now providing this correlation in an effort to bridge from the *in vitro* assessment using the electronic tongue setup (assessing the bitterness in buffer) to the *ex vivo* cleavage events (in saliva samples). By that approach we found, that the generated denatonium - as of the presence of proteases in saliva collected from patients - exceeded the limit of detection for denatonium of our tongues (10 nM) by a factor of 40 thereby providing evidence – in absence of direct testing in patients – that a strong taste is generated in peri-implant disease patients.

→ We are further now emphasizing that our sensor proofed the diagnostic concept in saliva samples from patients and that we did not test directly in patients. We are also justifying in the discussion (page 11) why we believe proof of concept was achieved by demonstrating the sensor's functionality in clinically obtained saliva samples *and* by the sensor-generated taste using the electronic tongue. We also emphasize (in the discussion and in the conclusion sections; page 11) that future clinical trials are needed to bridge from our approach in saliva samples to a clinical trial conducted in humans comparing the rates of taste sensations recorded in patients versus healthy volunteers. Lastly, we are now providing the correlation of the generated denatonium concentrations in saliva versus the bitterness of these concentrations as assessed by the electronic tongue setup.

In minor recommendation,

4. I think if you give **easy names to various linkers**, this manuscript would be more readable. 1st generation and 2nd generation and so many names in table (supplementary) are exist in manuscript.

5. Frankly, **1st generation linker may not be required in main manuscript**. I think it would be more interesting when sensor works are focused.

→ We are now detailing the respective sensor more thoroughly throughout the manuscript in its revised form. More importantly, we have updated Table S1 to facilitate the readers' orientation. We are now providing easy to read acronyms and we are using simplified color codes in the figures to improve the quality of the manuscript.

Reviewer #2 (Remarks to the Author):

6. The authors show that there is some level of non-specific reactivity (with either MMP9 or MMP1, depending on the substrate, see figure 3 and 4) but it is not clear if this will have an impact on the specificity of the assay to detect peri-implant disease. The authors only comment on the biomarker role of MMP8 in this clinical context, but I am not sure if the other MMPs are also implicated.

→We are now discussing this aspect (pages 3, 8 and 10 of the discussion) and detail that the sensor is not selective for an individual MMP but responding to sets of MMPs as present in inflammatory conditions including mucositis or peri-implantitis.

7. I think that an important piece is missing, as there is no clinical testing of the gum in patients. The authors went a long way to design the probe and link it to denatonium, to study the different degrees of "bitterness" of the cleaved compounds, and to formulate a chewing gum with the right amount of probes, and then did not show the effectiveness of the gum itself, but instead only used the alternative version of the probe (linked to fluorescein) to show that it is cleaved with saliva from patients.

→We agree with the reviewer and refer to our answer as given to questions #2 and #3.

→We apologize for our labeling error in Figure 6. We used a denatonium labeled probe but mislabeled the figure as if we had used fluorescein labels (it was correctly stated in the materials and methods and the SF section).

8. The head-to-head study in figure 6 is good, but it looks as if the probe is equally good as the standard test already used, so if there is not an improvement on the specificity/sensitivity the authors should at least comment on how they think a test like this could be implemented clinically and help patients. The idea of doing the test autonomously at home is proposed, but in practical terms how would this work?

→We are now discussing advantages of our chewing gum sensor which is detailed on page 9 and 10 in the revised manuscript.

Reviewer #3 (Remarks to the Author):

Title: Diagnosing peri-implant disease deploying the tongue as 24/7 detector

Authors: Ritzer et al.

The study of Ritzer et al. represents a new concept in diagnosing an oral condition referred to as peri-implant disease or peri-implantitis. This emerging disease is a condition that is increasing in all industrialized countries. Diagnosis and treatment of this condition is time-consuming and represents an increasing medical and economic burden. Early detection of this condition enables effective intervention and may contribute to the prevention of implant loss. The concept of self-diagnosis by using a sensory chewing gum is a novelty and will be greatly appreciated in the dental community.

The study comprises of a pre-clinical and a clinical part. The technical, i.e. the pre-clinical part of the study, has been described meticulously and has been tested in vitro. The device seems ready to be tested in humans.

Comments

9. The authors have tested their invention with saliva samples obtained from healthy subjects and patients with peri-implant disease, involving both mucositis and peri-implantitis. It would have been much better if they would have distinguished between mucositis and peri-implantitis. This would have been possible since the authors have selected patients with only mucositis and patients with peri-implantitis (Suppl. files, Clinical protocol).

→We are now providing the data for healthy volunteers, patients with mucositis and with peri-implantitis (Figure S15; discussed on page 10). Significant differences were observed for our sensor after incubation for 5 – 10 minutes in saliva for both, patients with mucositis or peri-implantitis as compared to control, respectively, and this is shown in comparison to the QuickZyme test and the chair-side test.

10. MMP-8 is a collagen-degrading enzyme from polymorphonuclear leucocytes. These leucocytes are present in the oral cavity in both health and disease which is also shown in Figure 6. Why did the authors select subjects with peri-implant inflammation for this study and not also gingivitis and periodontitis patients? All these conditions will provide a MMP-8-based positive signal with the new invention.

→We are now providing the outcome from periodontitis patients *versus* healthy volunteers. The sensor was unable to differentiate between patients with a reported periodontitis (n = 7) *versus* healthy volunteers (n = 7). None of the tests including our sensor was able to discriminate periodontitis patients from healthy volunteers. We are now detailing this outcome in Figure S16 as well as in the discussion section (page 8 of the revised manuscript). Regarding the gingivitis comment - our clinical protocol did not request a diagnosis of gingivitis which is why we cannot provide this data, retrospectively.

12. Why did the authors tested saliva samples and why not have subjects chew on the experimental device? The title suggests that the authors have tested “the tongue as a 24/7 detector” but this is not what actually has been studied. Also, the abstract suggest that subjects have actually used the device (40-42).

13. The authors did not show that the device works clinically.

As outlined in our responses to questions #2, 3, and 7 we could not perform these studies as of missing ethical approval to use our device in humans.

14. The corresponding author has filed several patents relating to diagnostic chewing gums (Patent number: 9526803, Publication number: 20150023879). This is not mentioned in the manuscript. One of these patents relates to oral inflammation in general and not to peri-implant disease in particular. Such a patent can be related to a conflict of interest.

→Thank you for this valuable comment. Accordingly, we are now stating a possible conflict of interest and mention the patent application, accordingly.

Reviewers' Comments:

Reviewer #1:

Remarks to the Author:

As I recommended, this sensor looks depend on individual tongue capability. Even though electronic tongue tests were performed, the clinical experiments would be required because quantification of bitterness is required by patient's real tongue. I think it is the core in this idea. As author claimed, the proof of this diagnosis concept is required.

Reviewer #2:

Remarks to the Author:

The authors addressed all of the comments to the best of their abilities with their current resources. The manuscript now is sound in my opinion though it lacks clinical validation. The work may spur additional interest to further the studies in patients.

Reviewer #3:

Remarks to the Author:

The authors have made a serious attempt to comply with the comments of this reveiwer. Additonal testing has been performed. The missing part of the study however remains a true clinical trial with patient's response as the study outcome.

It is also surprising that the sensor is not able to discriminate between periodontal health and periodontal disease, whereas it does between saliva from mucositis and saliva from peri-implantitis patients.

Future clinical trials are needed to validate the results of the present study.

I would agree to accept the manuscript for publication in the present form.

On behalf of all authors, I like to express our great appreciation for the valuable comments made by the reviewers. We are providing the reviewers' feedback in the frame, followed by our comment.

Reviewer #1 (Remarks to the Author):

As I recommended, this sensor looks depend on individual tongue capability. Even though electronic tongue tests were performed, the clinical experiments would be required because quantification of bitterness is required by patient's real tongue. I think it is the core in this idea. As author claimed, the proof of this diagnosis concept is required.

Reviewer #2 (Remarks to the Author):

The authors addressed all of the comments to the best of their abilities with their current resources. The manuscript now is sound in my opinion though it lacks clinical validation. The work may spur additional interest to further the studies in patients.

Reviewer #3 (Remarks to the Author):

The authors have made a serious attempt to comply with the comments of this reviewer. Additional testing has been performed. The missing part of the study however remains a true clinical trial with patient's response as the study outcome. It is also surprising that the sensor is not able to discriminate between periodontal health and periodontal disease, whereas it does between saliva from mucositis and saliva from peri-implantitis patients. Future clinical trials are needed to validate the results of the present study. I would agree to accept the manuscript for publication in the present form.

We initially defined the study on saliva samples collected from patients and healthy volunteers as a 'clinical study' or 'clinical trial'. The reviewers' comments indicated a different interpretation of these terms in that a clinical study or trial should include patient reported outcomes on taste. Therefore, we have carefully revised the entire manuscript to adopt the reviewers' interpretation. To this end, we have strengthened our initial statement that studies with patient reported outcome are required in the future by providing an additional statement at the end of the discussion section (p9) stating: "The study lacks a clinical trial at this stage". We further removed the terms 'clinical study' or 'clinical trial' from the keywords and changed the abstract by removing "clinical" from the following sentence: "The peptide sensors proofed significant ~~clinical~~ success". Similar changes were made on p4 (end of the introduction section), p8 (Heading of the materials section), two times on p9 (beginning of the discussion section) as well as on p11 (end of the discussion section). Finally, the figure legend of Fig. 6 was changed, accordingly.